# Distribution and risk assessment of pesticide residues in sediment samples from river Ganga, India

**Zeshan Umar Shah** [ID]**\*, Saltanat Parveen** [ID]

Department of Zoology, Limnology Research Laboratory, Aligarh Muslim University, Aligarh, India

\* zeshanomer11@gmail.com

**Data Availability Statement:** All relevant data are within the paper and its Supporting Information files.

**Funding:** The authors received no specific funding for this work.

## Abstract

Indiscriminate use of pesticides leads to their entry in to the bottom sediments, where they are absorbed in the sediment's particle and thus, may become the consistent source of aquatic pollution. The present work was carried out to evaluate pesticide residues in the sediment samples and associated human health risk of commonly used pesticides along the basin of river Ganga. Total of 16 pesticides were analyzed along three stretches of river Ganga. The concentration of pesticides in the upper stretch ranged from ND to 0.103 μg/kg, in the middle stretch ND to 0.112 μg/kg, and in the lower stretch ND to 0.105 μg/kg. Strong positive correlation was found between total organic carbon and total pesticide residues in sediment samples. Carcinogenic and non-carcinogenic values were estimated below the threshold limit suggesting no associated risk. Risks associated with the inhalation route of exposure were found to be higher than the dermal and ingestion routes. Children were found at higher risk at each site from multiple routes of exposure than adult population groups. Toxic unit values were found to be below the threshold value suggesting no risk associated with exposure of pesticides from sediments. However, long term effects on ecological quality due to consistent pesticide exposure must not be ignored. Therefore, the present study focuses on concrete efforts like lowering the irrational used of pesticides, tapping of agricultural and domestic drains, advice to farmers for appropriate use of pesticide doses, to reduce the threat of pesticide pollution in the river system and possible human health risk.

## Introduction

Pesticides belong to the class of chemicals substances used to improve and increase crop production. However after use their residues tend to persist and accumulate in the different environmental compartments. In the aquatic ecosystem the rate of bio-accumulation depends on the solubility and octanol-water partition coefficient (logKow) of the pesticides. The residues present in the environment have impact on both environment and human health even at low concentration [1–5]. Furthermore, physical, chemical and biological agents of the environment degrade the parent compound into few or more components that become more

**Competing interests:** The authors have declared that no competing interests.

persistent and toxic to organisms [6–9]. Various studies have shown incidence of various cancers in humans [10–12] including teratogenicity [13,14], endocrine dysfunctioning [15,16], nerve dysfunctioning [17,18], and genotoxicity [19,20]. Similarly, wildlife organisms on exposure to pesticides have shown developmental deformities in genitalia [21], abnormal reproductive behavior [22], sterility, cancers, egg-shell thinning, and immune dysfunctioning [23,24].

The potential ecotoxicological risks associated with pesticides are addressed with determination of Toxic units (TUs) and risk quotients (RQs) [1]. Large numbers of studies considering the above criteria have been carried out in many countries [7,25–27] including India [2,28–31]. The pesticide residues also tend to accumulate in the sediments of the aquatic ecosystem by their absorption and thus, may become a consistent source of aquatic pollution. Therefore based on the guidelines of EC 2000, EC 2008, EU 2013, it is undoubtedly needed to include pesticide concentration in sediment for risk assessment. Although a variety of discussions and methods are available on sediment risk assessment associated with pesticide toxicity, these are rarely applied [28,32–34].

The other problem usually discussed is the impact by mixture of different pesticides present at a time. Thus, there is a need to establish the real impact of these mixtures (pesticides) on biota communities of the ecosystem [35–37], which can be predicted by the independent action of the pesticide or by concentration addition impact. Consumers assume the different action of pesticides, therefore ignoring the agonistic/antagonistic action of mixture and overestimating the effect [38–40]. Concentration addition is often the recommended first step towards impact process as it provides the worst case scenario of mixture effect [41].

River Ganga is the largest and most affected river of India as large amounts of domestic and agricultural wastes runoff into the river. This raises a serious concern on concentrations of the cocktail of pesticides present in water [2]. The rising concentration of pesticide residues in aquatic ecosystems leads to degradation and significant biodiversity loss [42]. In addition, human exposure to contaminated water via bath/swimming and ingestion poses serious risk [43,44]. Therefore there is an increase in assessing and predicting the environmental risk of pesticides by employing ecotoxicological indices [1,45–50]. Risk assessment involves estimation of the possible impact posed on non-target organisms upon environmental pesticide concentration exposure. The toxic unit (TUs) estimation is the deterministic approach towards the ecological risk assessment of pesticides to non-target aquatic organisms. Toxic unit for sediment samples is determined by using the individual concentration of pesticide and its critical toxicological endpoint for the given organism [51].

Risk assessment involves assessment of potential risk posed to humans upon exposure to environmental contaminants. Estimating the risk of pesticides depends on the route of exposure to human body as contaminants may occur in different media sources of the aquatic ecosystem [52–58]. The exposure pathway to pesticides includes both dietary and non-dietary exposures [59–61]. Dietary exposure route involves exposure to contaminants via food and water contaminated with pesticide residues while non-dietary exposure involves dermal contact with contaminated water and sediments while bathing/ swimming, inhalation of residues from sediments and probable ingestion of contaminated sediments [57,62,63].

The objective of this study was to establish the concentration of pesticide residues in sediment samples along three different stretches of river Ganga and based on these concentrations ecological risk on non-target organisms using deterministic approach Toxic Units (TUs) and determination of carcinogenic and non-carcinogenic risk via different exposure routes into the human body. Thus, the study is important as it mainly focuses on the deterministic and probabilistic approach towards risk assessment of pesticides pollution in sediments. Furthermore, the river plays an important socio-economic role in the development of the country as it is an essential habitat for diverse fishes and provides other ecosystem services.

## Materials and methods

### Sampling site and collection of sediments

Three stations were selected along the whole length of river Ganga for pesticide analysis in the sediment samples. In the upper stretch Rishikesh station (30°03′51″N, 78°17′28″E), in the middle stretch Narora station (28°12′02″N, 78°23′41″E) and in the lower stretch Patna station (25° 36′28″N, 85° 08′34″E). The river Ganga originates from Gangotri in the Himalayas and flows through highly populated agricultural areas of Uttarakhand, Uttar Pradesh and Bihar, the deposits of organic pollutants are drifted through surface runoff into the river water. The most cultivable vegetation includes paddy, wheat, sugar cane etc.

A total of 40 random samples from bottom (3–4 meters) sediments were collected using Ekman sediment grabber operated manually from the bridge and placed in aluminium foil [64]. For further analysis, an ice box was used to store immediately and transport all samples to the laboratory.

### Chemicals and reagents

Organic solvent as n-hexane and acetone (HPLC grade) were purchased from Sigma Aldrich. Co., USA and were glass distilled before use. Anhydrous Sodium Sulphate ($Na_2So_4$) and Sulphuric acid ($H_2So_4$) were procured from Himedia Pvt. Ltd. New Delhi, India. Pesticides of purity (>99%) were purchased from Sigma Aldrich, USA and Rankem Pvt. Ltd. New Delhi, India.

### Extraction of samples

Pesticide residues in the sediment samples were extracted by Soxhlet extraction method [65]. 100 g of sediment of each sample was packed in blotting paper and transferred into a thimble fitted with a condenser and connected with a round bottom flask containing 150 ml of (1:1) hexane and acetone solvent. Solvent flask was refluxed for 8 hours at 85°C. The extract was finally concentrated on a water bath to 1 ml and redissolved in 10ml of distilled water and cleaned up by solid phase extraction method with Oasis HLB cartridges. Before use the cartridges were conditioned to 10 ml of (1:1) n-hexane and acetone solvent. The concentrated samples were percolated through cartridges at a flow rate of 5 ml/min. The cartridges were then rinsed with 5 ml of ultrapure water and vacuum dried to remove excess of water. Finally the retained pesticides were eluted with another 6 m of (1:1) n-hexane and acetone solvent and were collected in glass tubes. The eluted samples were then reduced to 1 ml on water bath at 50°C.

### Analysis of samples

The equipment used in the analysis of pesticides in sediment samples was liquid chromatography equipped with a mass spectrometer (TripleTof 5600+Make Sciex). Cleaned extracts were used for analysis of pesticide residues. C18 column (2.1 mm × 100 mm, 5 μm particle size) was used as the analytical capillary column. A flow rate of the eluent was maintained at 0.2 ml/min for 10 minutes. The mobile phase consisted of solutions A (5 mM $CH_3COONH_4$ in water) and B (5 mM $CH_3COONH_4$ in 1: 1 (n-hexane-acetone)). Thermostat temperature was maintained at 35°C and injection volume was 10 μL. Presence of the quantification, confirmation ions and retention time was based on the authentic standards (Sigma Aldrich).

### Quality control and quality assurance

Analytes used were subjected to rigorous quality control and quality assurance methods for reliability of the results. HPLC grade reagents and double distilled deionized water was used throughout the experiment. Blank was always run for the correction of reading of the

instrument. Internal standard ensures the accuracy of the extract and cleaning method. Limit of detection (LOD), limit of quantification (LOQ) and recovery rate was within the range of 73 to 90% (S1 Table), peaks were identified by comparing internal standards with extract retention time.

## Statistical analysis

The results of the pesticide analysis obtained were presented as mean and standard deviation using Microsoft excel 2010 and visualized by Box and whisker plot. Significant differences in concentration of pesticide residues were evaluated by using one way ANOVA (Microsoft excel 2010) and Duncan's Multiple Range Test (DMRT) (SPSS 16.0).

## PCA analysis

Pesticides (organochlorine, organophosphate, carbamate) detected in sediment samples were subjected to Principal Component Analysis (PCA) using SPSS 16 to infer hypothetical sources of pesticides. PCA was performed with varimax rotation because it minimizes the number of variables with high loading on each component, thus facilitating the interpretation of PCA results [66].

## Health risk estimation

In estimating the human health risk of pesticide residues observed in sediment samples of the three stretches of river Ganga, three possible routes of exposure were considered. The routes are: ingestion of contaminated sediments, dermal contact with contaminated sediments, and inhalation of residues from contaminated sediments. Specific risk assessment models have been used for this purpose.

## Toxic units

The toxic units were used for the estimation of risk of pesticides in sediment samples for algae, daphnia and fish based on the method of [1]. As per regulatory framework for chemicals (REACH) the toxicity tests for chemicals require different trophic organisms (primary producers, primary and secondary consumers) to protect more sensitive aquatic community.

For the estimation of the toxic unit associated with sediment samples pore water concentration was estimated in the equilibrium-partitioning approach. This is the important approach towards establishing sediment toxicity benchmarks [67]. Pore water concentration ($C_{pw}$) in the sediment samples was estimated following the formula given below.

$$Cpw = \frac{CS}{Kd} \tag{1}$$

Where Kd is partitioning coefficient, Cs is the sediment concentration, Cpw is pore water concentration of the pesticides.

Kd was estimated by the given formula

$$Kd = Koc \times foc$$

Where, Koc is the organic carbon water partitioning coefficient for the pesticides.

Foc is a fraction of the total organic carbon measured in the sediment samples.

Log Koc is the octanol-water partitioning coefficient and is determined by the formula given below.

$$Log\ Koc = a \times Log\ Kow + b$$

The constant (a) and (b) were set to be 0.72 and 0.49 [68].

## Chronic daily intake (CDI)

The United States Environmental Protection Agency (USEPA, 1997) has developed a model to estimate the non-carcinogenic risks of pesticide exposure to contaminants in adults and children through dietary and non-dietary exposure. Human exposure to the pesticides can occur either through ingestion, through bathing or recreational activities.

River Ganga is the place where large numbers of devotees all around India are attracted for religious activities who come to take bath or swim. In addition a large percentage of water is used for irrigation and household usage. The formula given below was used to estimate non-carcinogenic risks.

$$Hazard\ quotient\ (HQ) = \frac{CDI}{RfD} \tag{2}$$

The carcinogenic risk for each pesticide was estimated using the cancer slope factor. The following formula was used.

$$R = CDI \times SF \tag{3}$$

Where R is carcinogenic risk, CDI is the chronic daily intake, whereas RfD and SF are individual pesticides reference dose and slope factor.

All the routes of exposure (dermal contact, ingestion and inhalation) were subjected to estimation of non-carcinogenic risk in adults and children. Exposure due to inhalation may occur when particular matter (dust), vapors or aerosols containing contaminant residues are released during low tides from dried sediments, exposure of dermal contact may occur by taking bath or swimming in the river, ingestion of pesticides may occur by drinking the water of eating food from the river (USEPA, 2017). For each population group formula 4,5,6 used to estimate the CDI of contaminated sediment samples was taken from [69]. The input parameters used are presented in (S3 and S4 Tables).

$$CDI\ ingestion = \frac{C\ (sediment) \times IR\ (sediment) \times CF \times EF \times ED}{BW \times AT} \tag{4}$$

$$CDI\ inhalation = \frac{C\ (sediment) \times \left(\frac{1}{PET}\right) \times IAR \times EF \times ED}{BW \times AT} \tag{5}$$

$$CDI\ dermal = \frac{C\ (sediment) \times SA \times CF \times EF \times ED \times ABS \times AF}{BW \times AT} \tag{6}$$

Where,

C = Concentration of pesticide in sediment ($\mu g\ kg^{-1}$), IR = Ingestion rate, CF = Conversion factor, EF = Exposure frequency, ED = Exposure duration, BW = Body weight, AT = Average life span, PEF = Particle emission factor, IAR = Inhalation rate, SA = Surface area, ABS = Dermal absorption factor, AF = Dermal surface factor.

## Results and discussion

Contamination of sediment with pesticides is a common feature of the riverine system that drains large areas of intense agricultural fields. Sediment analysis of river Ganga showed a diverse classes of pesticides at varying concentrations. The sediment samples collected from three different stations were analyzed for 15 pesticides. In the upper stretch (Fig 1) highest mean concentration was methylparathion (0.103 µg/kg dry weight (dw)), while the lowest mean concentration being nuarimol (0.056 µg/kg dw). Chlordane, heptachlor, methoxychlor, dichlorvos and dimethoate were not found in the sediment samples. The concentration of other pesticides was malathion (0.096 µg/kg dw), azinphosmethyl (0.084 µg/kg dw), tridemorph (0.070 µg/kg dw) and cypermethrin (0.063 µg/kg dw).

Maximum number of pesticides was found at Narora station (Fig 2) of middle stretch with highest mean concentration being cypermethrin (0.112 µg/kg dw), while the lowest mean concentration being nuarimol (0.062 µg/kg dw). The most ubiquitous pesticides were tridemorph (0.107 µg/kg dw), dichlorvos (0.099 µg/kg dw), chlordane (0.098 µg/kg dw), atrazine (0.096 µg/kg dw) and malathion (0.093 µg/kg dw).

In the lower stretch (Fig 3) highest mean concentration being chlordane (0.105 µg/kg dw), while the lowest mean concentration being methylparathion (0.073 µg/kg dw). Atrazine was not detected in the analyzed samples. The concentration of other pesticides was methoxychlor 0.103 µg/kg dw, dichlorvos 0.102 µg/kg dw, cypermethrin 0.096 µg/kg dw, azinphosmethyl 0.093 µg/kg dw, tridemorph 0.092 µg/kg dw, binapacryl 0.091 µg/kg dw and nuarimol 0.084 µg/kg dw.

The detected pesticides have high octanol/water partition coefficient (log $k_{ow}$), hydrophobic, low water solubility and therefore tend to accumulate more in the sediments. Other factors including time and rate of consumption are also important in the accumulation of pesticides. The total usage of pesticides in Ganga basin between years 2017–2019 was 72,741 MT, which is 27% of countries total consumption [70]. Pesticide usage in the basin shows increase along

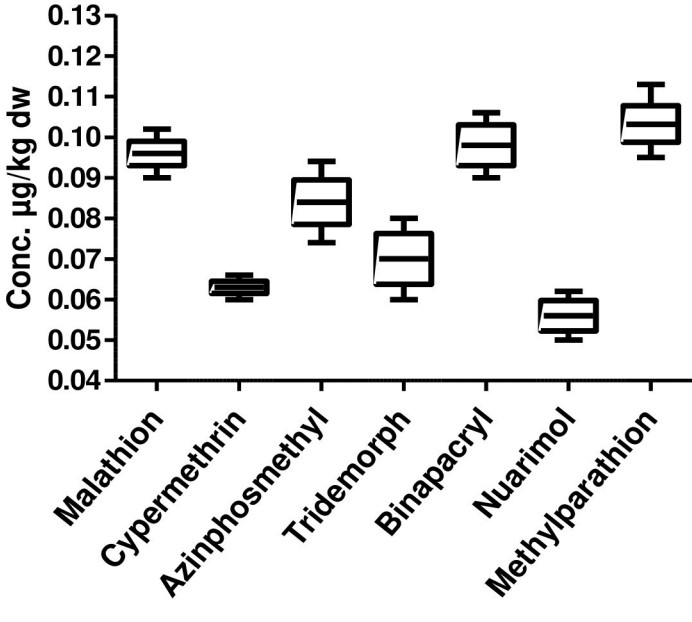

**Fig 1. Box and whisker's plot of pesticide residues in sediment samples from upper stretch of river Ganga.**

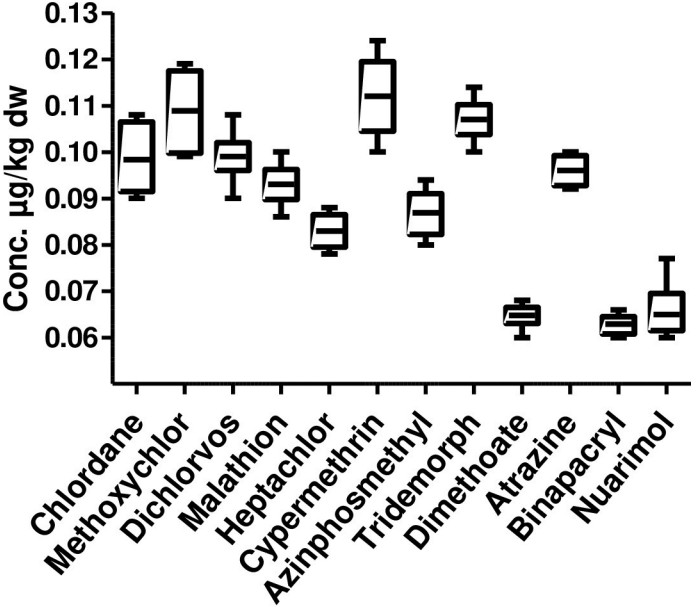

**Fig 2. Box and whisker's plot of pesticide residues in sediment samples from middle stretch of river Ganga.**

the years. Thus enormous quantity of pesticide usage significantly contributes to bioaccumulation of their residues in the ecosystem through several means. Azinphosmethyl, dimethoate, atrazine, dichlorvos, cypermethrin, chlormequat, tridemorph, fenobucarb etc., are

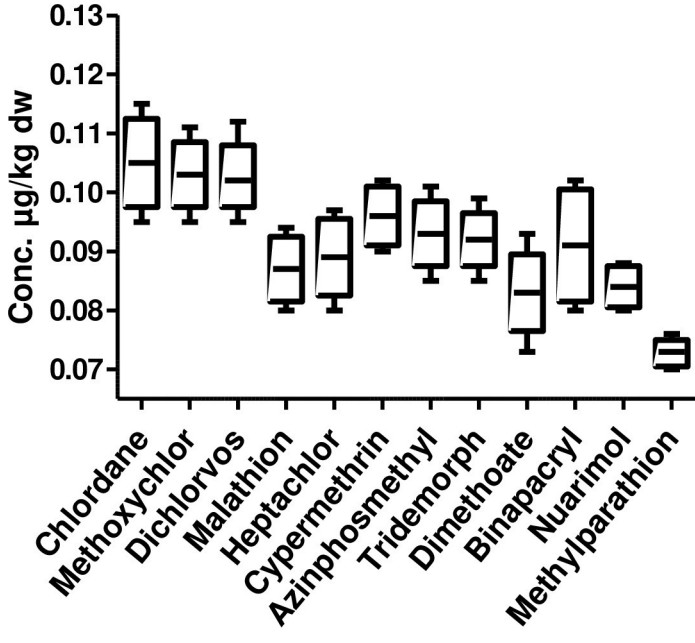

**Fig 3. Box and whisker's plot of pesticide residues in sediment samples from lower stretch of river Ganga.**

continuously used in along the basin in abundant quantities [71]. Detection of malathion, heptachlor and chlordane shows their historical use and persistence in the ecosystem [72]. The findings in our study are in partial accordance with the finding of [4] who have observed lindane, methylparathion, endosulfan and DDT from Champanala, Mond ghat and Burning ghat at Bhagalpur station of river Ganga.Malik *et al*. [73] have observed similar types of pesticides heptachlor (3.85 ng/g) and methoxychlor (0.47 ng/g) from sediment samples of Gomti, a tributary of river Ganga. Mondal *et al*. [28] have observed a class of persistent organochlorine and organophosphate pesticides from sediment analysis of Hooghly River in West Bengal.

Majority of chemical pollutants discharged into aquatic resources end up concentrated in sediments that act as sink as well as source of the toxicity. The results of this study highly contribute to the fact that sediments play an important role in retention of pesticides in the aquatic ecosystem. Large proportion of available pesticides bind to suspended particles and settle down at the bottom of the river [74,75]. Presence of pesticide residues in the sediments can inhibit microbial processes thus affecting the degradation rate of organic matter, which could result in generation of anaerobic gasses that affect growth of ecosystem [76]. There was a positive correlation between total organic carbon and pesticide concentration in sediments, which indicates that sediment organic carbon could enhance adsorption and deposition of pesticide residues because of their hydrophobicity (Fig 4). The total organic carbon percentage in the sediments collected from three stations vary, in the upper stretch mean percentage was 1.67%. In the middle and lower stations mean percentage were 1.88 and 1.89%. Therefore higher amounts of TOC were mentioned in middle and lower stretches, thus pesticide deposition was higher than upper stretch. The increase of organic matter content in soil can supply more

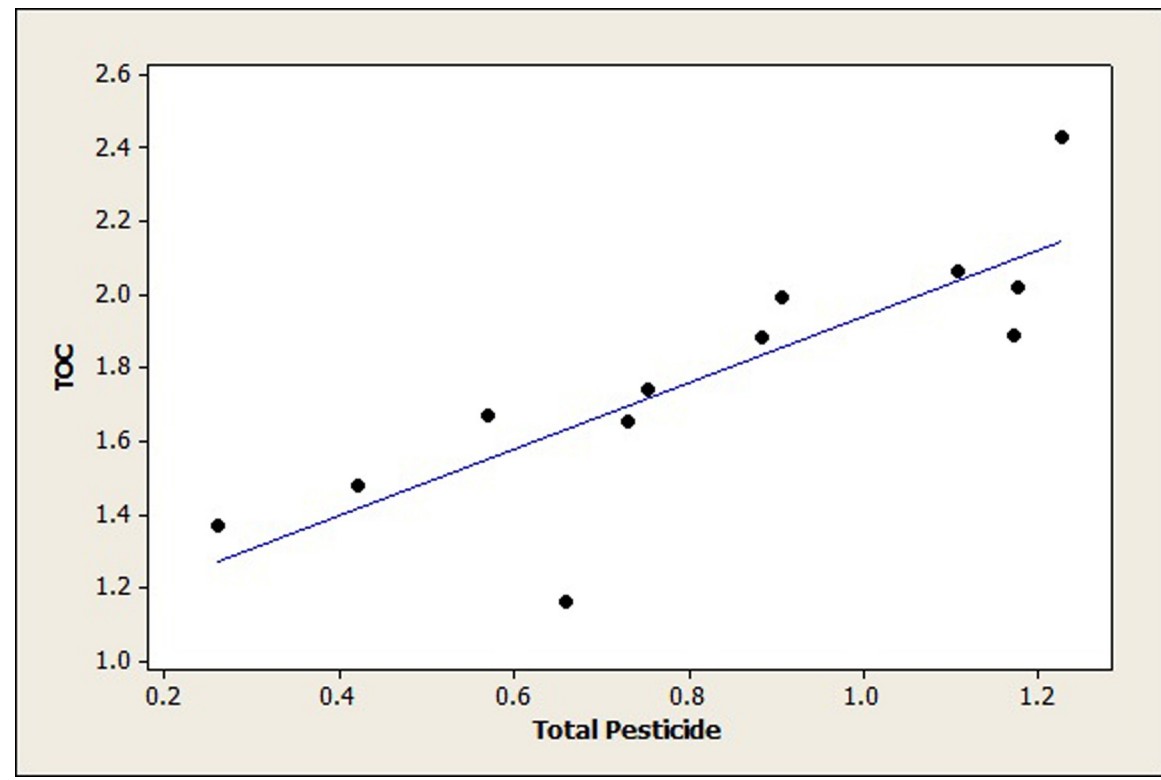

**Fig 4. Correlation between total organic carbon and total pesticide concentration in sediment samples collected from three stations of river Ganga ($r^2$ = 0.8).**

carbon source to facilitate microbial degradation of pesticide residues [77]. As a result, the content of TOC could have an impact on the residues of pesticides in soil [78,79]. These results are similar to the findings of [80] who reported significant correlation between pesticide and total organic carbon from surface soil collected from three major states of north eastern parts of India. Ogbeide *et al.* [47] also reported positive correlation between total organic carbon and pesticide concentration in sediments collected from Owan River, Edo State Nigeria. It can be suggested that the high contribution of pesticides in sediments has probably originated from a similar contribution source. It has been already reported that the higher the value of organic carbon higher the partition coefficients ($K_{OC}$), suggesting that these compounds get strongly adsorbed in the sediments [81].

Pesticide residue levels found in this study were compared with various studies from the world (S2 Table). The observed concentrations in this study were lower than the several studies from China [82–84] and Niger river Nigeria [85]. While these fall in the range of those observed from Ikpoda River and Ebro River from Nigeria [86] and Spain [1]. The concentrations were above the levels observed from the urban river in Florida and four urban creeks in California, USA [64,87] and Lake Tashk, Iran [88] (S2 Table).

Principal component analysis of sediment samples is presented in (Fig 5). The figure shows two components that account for 75.90% of the total variance. PC1 is explained by positive

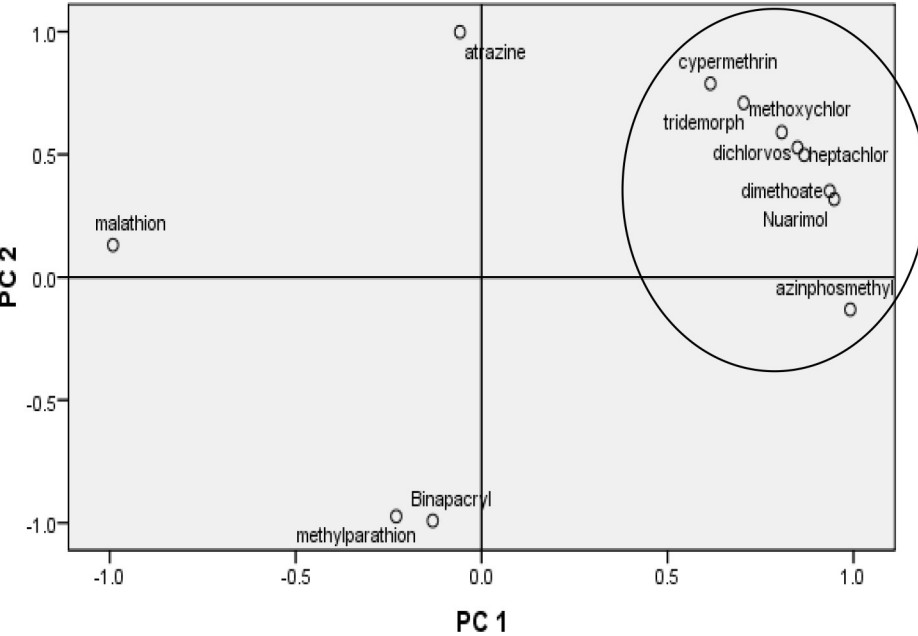

**Fig 5. Principal component plot of pesticide residues in sediment samples from river Ganga.** The determination of carcinogenic and non-carcinogenic risk associated with pesticide exposure from sediments includes three routes. The non-carcinogenic risk estimation from the three stations (Tables 1, 2 and 3) were below the reference levels (USEPA) for each studied pesticide, therefore these contaminants are unlikely considered to pose potential risk to human health [69]. According to the USEPA, guidelines if the estimated CDI value of pesticide exceeds oral reference dose (S3 Table, USEPA) value is supposed to pose adverse health impact or non-carcinogenic risk to human health. The estimated CDI value for each pesticide showed that children are at higher risk than adults. Ogbeide *et al.* [34] have reported similar results for CDI values from Illushi River, Ogbesse River and Owan River of Nigeria. Carcinogenic risk from the exposure of pesticides in sediments was below the threshold limit of $10^{-6}$ (USEPA) indicating no observed risk for both adults and children population groups at each station. Estimated results for non-carcinogenic and carcinogenic risk showed, pesticide exposure from the inhalation route has higher potential to cause risk in adults and children than by other two routes. Further, the study also clarifies that there is a higher potential of cancer risk through multiple exposure routes in children than in adult population groups.

**Table 1. Estimation of Chronic daily intake (CDI) and hazard quotient (HQ) for pesticide residues in sediment from middle stretch.**

| | Non-carcinogenic | | | | | | Carcinogenic | | | | | |
|---|---|---|---|---|---|---|---|---|---|---|---|---|
| | Adult | | | Child | | | Adult | | | Child | | |
| | Ingestion | Inhalation | Dermal | Ingestion | Inhalation | Dermal | Ingestion | Inhalation | Dermal | Ingestion | Inhalation | Dermal |
| Chlordane | 2.23E-06 | 3.33E-014 | 1.43E-03 | 1.00E-05 | 1.78E-011 | 3.73E-03 | 4.69E-01 | 7.0E-019 | 3.07E-08 | 2.11E-010 | 3.74E-016 | 7.84E-08 |
| Methoxychlor | 2.98E-08 | 4.0E-016 | 1.91E-05 | 1.351E-07 | 2.39E-013 | 4.99E-05 | | | | | | |
| Dichlorvos | 2.70E-07 | 4.00E-015 | 1.74E-04 | 1.23E-06 | 2.10E-012 | 4.52E-04 | 3.91E-011 | 5.80E-015 | 2.52E-08 | 1.79E-010 | 3.04E-016 | 6.55E-08 |
| Malathion | 6.36E-010 | 1.12E-016 | 4.09E-06 | 2.89E-08 | 5.10E-014 | 1.06E-05 | 4.87E-014 | 7.61E-021 | 3.10E-010 | 2.19E-012 | 3.87E-018 | 8.09E-010 |
| Heptachlor | 2.26E-08 | 4.00E-015 | 1.46E-04 | 1.02E-06 | 1.82E-012 | 3.80E-04 | 5.08E-010 | 9.00E-018 | 3.28E-07 | 2.31E-09 | 4.09E-015 | 8.55E-07 |
| Cypermethrin | 1.53E-08 | 2.01E-016 | 9.85E-06 | 2.29E-03 | 1.23E-013 | 2.56E-05 | | | | | | |
| Azinphosmethyl | 5.95E-010 | 1.01E-017 | 3.82E-07 | 2.69E-09 | 4.77E-015 | 9.95E-07 | | | | | | |
| Tridemorph | 1.46E-08 | 2.01E-016 | 9.41E-06 | 6.65E-08 | 1.17E-014 | 2.44E-05 | | | | | | |
| Dimethoate | 4.38E-07 | 5.01E-015 | 2.81E-04 | 1.97E-06 | 3.50E-012 | 7.30E-04 | | | | | | |
| Atrazine | 3.74E-09 | 5.71E-017 | 2.41E-06 | 1.70E-08 | 3.01E-014 | 1.10E-05 | 2.88E-011 | 4.40E-019 | 1.85E-08 | 1.31E-010 | 2.31E-016 | 8.53E-08 |
| Nuarimol | 1.41E-08 | 1.66E-016 | 9.08E-06 | 6.41E-08 | 6.38E-08 | 2.35E-05 | | | | | | |

**Table 2. Estimation of Chronic daily intake (CDI) and hazard quotient (HQ) for pesticide residues in sediment from Upper stretch.**

| | Non-carcinogenic | | | | | | Carcinogenic | | | | | |
|---|---|---|---|---|---|---|---|---|---|---|---|---|
| | Adult | | | Child | | | Adult | | | Child | | |
| | Ingestion | Inhalation | Dermal | Ingestion | Inhalation | Dermal | Ingestion | Inhalation | Dermal | Ingestion | Inhalation | Dermal |
| Malathion | 6.55E-009 | 1.01E-016 | 4.22E-06 | 2.98E-08 | 5.26E-014 | 1.09E-05 | 4.97E-013 | 7.61E-021 | 3.20E-010 | 2.27E-012 | 4.00E-018 | 8.35E-010 |
| Heptachlor | | | | | | | | | | | | |
| Cypermethrin | 8.63E-09 | 1.01E-016 | 5.54E-06 | 3.92E-08 | 1.27E-09 | 1.44E-05 | | | | | | |
| Azinphosmethyl | 5.75E-010 | 1.01E-013 | 3.69E-07 | 2.61E-09 | 3.69E-07 | 9.61E-07 | | | | | | |
| Tridemorph | 9.58E-09 | 1.01E-016 | 6.15E-06 | 4.35E-08 | 7.67E-014 | 1.60E-05 | | | | | | |
| Nuarimol | 1.31E-08 | 1.66E-016 | 8.20E-06 | 5.78E-08 | 1.02E-013 | 2.13E-05 | | | | | | |
| Methyl parathion | 5.64E-07 | 8.01E-015 | 3.62E-04 | 2.55E-06 | 4.51E-012 | 9.40E-04 | | | | | | |

**Table 3. Estimation of Chronic daily intake (CDI) and hazard quotient (HQ) for pesticide residues in sediment from lower stretch.**

| | Non-carcinogenic | | | | | | Carcinogenic | | | | | |
|---|---|---|---|---|---|---|---|---|---|---|---|---|
| | Adult | | | Child | | | Adult | | | Child | | |
| | Ingestion | Inhalation | Dermal | Ingestion | Inhalation | Dermal | Ingestion | Inhalation | Dermal | Ingestion | Inhalation | Dermal |
| Chlordane | 2.38E-06 | 3.33E-014 | 1.53E-03 | 1.08E-05 | 1.91E-011 | 3.81E-03 | 5.00E-011 | 7.0E-019 | 3.23E-08 | 2.27E-010 | 4.02E-016 | 8.01E-08 |
| Methoxychlor | 2.82E-08 | 4.0E-016 | 1.81E-05 | 1.28E-07 | 2.25E-013 | 4.70E-05 | | | | | | |
| Dichlorvos | 2.78E-08 | 4.00E-015 | 1.02E-04 | 1.26E-06 | 2.24E-012 | 4.66E-04 | 4.03E-012 | 5.80E-019 | 1.48E-08 | 1.83E-010 | 3.24E-016 | 6.75E-08 |
| Malathion | 5.95E-09 | 1.0E-016 | 3.82E-06 | 2.69E-08 | 4.77E-014 | 9.95E-06 | 4.52E-013 | 7.60E-021 | 2.90E-010 | 2.04E-012 | 3.62E-018 | 7.56E-010 |
| Heptachlor | 2.42E-07 | 4.00E-015 | 1.56E-04 | 1.10E-06 | 1.95E-012 | 4.06E-04 | 5.44E-010 | 9.00E-018 | 3.51E-07 | 2.47E-09 | 4.39E-015 | 9.13E-07 |
| Cypermethrin | 1.31E-08 | 1.01E-016 | 8.44E-06 | 5.96E-09 | 1.05E-013 | 2.19E-05 | | | | | | |
| Azinphosmethyl | 6.35E-010 | 1.01E-017 | 4.09E-07 | 2.89E-09 | 5.09E-014 | 1.06E-06 | | | | | | |
| Tridemorph | 1.26E-08 | 2.01E-016 | 8.09E-06 | 5.72E-08 | 1.00E-013 | 2.10E-05 | | | | | | |
| Dimethoate | 5.65E-07 | 1.01E-014 | 3.65E-04 | 2.58E-06 | 4.50E-012 | 9.50E-04 | | | | | | |
| Nuarimol | 1.96E-08 | 3.33E-016 | 1.23E-05 | 8.66E-08 | 1.53E-013 | 3.20E-05 | | | | | | |
| Methyl parathion | 4.01E-07 | 2.88E-015 | 2.56E-04 | 1.81E-06 | 9.98E-021 | 6.68E-04 | | | | | | |

correlation loading for the variables: heptachlor, dichlorvos, dimethoate, nuarimol, tride-morph, methoxychlor and cypermethrin. PC2 is positively loaded for malathion and atrazine. The positive loading of malathion in PC2 indicates that it has different degradation and distribution pattern. Positive loading of pesticide residues in a component implies that these residues undergo a similar degradation and distribution pattern in the medium [32,47,89].

The toxic unit (TU) is used to estimate the potential ecological risk of pesticide residue in sediments (Table 4). The observed toxic units for sediments were <1 which indicates that there is no potential acute risk with presence of pesticide residues in sediments. The observed results in this study are in accordance with [1] and [86] who reported sediment toxic units <1 for pesticides in Ikpoda river from Nigeria and Ebro River from Spain. The observed results are below the threshold value, which suggest no acute effect due to the presence of pesticides. However complex chronic effects and potential loss of ecological quality cannot be ignored. The pattern of pesticide pollution, including its impact on species, could be attributed to the recent decline in abundance of species diversity as reported by Sarkar [90].

## Conclusion

The study carried out aimed to provide information regarding the distribution and risk of commonly used pesticides along the basin of river Ganga. Observed concentrations of studied pesticides along the three stations may have potential of chronic effect to both the aquatic species and associated humans. Standard protocol developed for carcinogenic and non-carcinogenic risk assessment is considered as tool to give the probable risk associated with pesticide exposure from sediments. The estimated results showed no acute risk of cancer and non-cancer effects from multiple routes (ingestion, dermal contact and inhalation) of exposure to pesticides present in sediments to both adult and children groups of populations. Probabilistic results showed pesticide exposure from inhalation would have higher risk to both population groups, nevertheless children are at higher risk through multiple routes of exposure than adults. Probable ecological risk assessment was determined by toxic unit values and was found to have a potential risk to the aquatic organisms inhabiting the ecosystem. Results from this study highlight the need of concrete steps to be taken for regular monitoring and pesticide pollution control along the river basin, thus, to reduce the associated risk on aquatic ecosystems and associated humans.

**Table 4. Risk assessment of pesticide residues in sediment samples based on toxic units calculation.**

| Pesticide | Middle stretch | Upper stretch | Lower stretch |
|---|---|---|---|
| Chlordane | 1.25E-02 | | 1.10E-02 |
| Methoxychlor | 1.39E-02 | | 1.32E-02 |
| Dichlorvos | 6.75E-02 | | 6.99E-02 |
| Malathion | 2.24E-02 | 2.62E-02 | 2.11E-02 |
| Heptachlor | 8.99E-03 | | 9.69E-03 |
| Cypermethrin | 1.13E-02 | 7.19E-03 | 9.74E-03 |
| Azinphosmethyl | 1.86E-02 | 2.03E-02 | 2.00E-02 |
| Tridemorph | 1.79E-02 | 1.32E-02 | 1.55E-02 |
| Dimethoate | 3.22E-02 | | 4.19E-02 |
| Atrazine | 2.14E-02 | | |
| Binapacryl | 8.52E-03 | 1.50E-02 | 1.23E-02 |
| Nuarimol | 1.18E-02 | 1.20E-02 | 1.60E-02 |
| Methyl parathion | | 2.41E-02 | 5.38E-02 |

## Supporting information

**S1 Table. Limit of detection (LOD), limit of quantification (LOQ) and recovery percentage of sediments fortified with 10 μg/kg (n = 10) of pesticides in sediment.**
(PDF)

**S2 Table. Few of the international studies showing concentration of observed pesticides.**
(PDF)

**S3 Table. Represents the estimated input parameters used in calculation of chronic daily intake (USEPA, 2017).**
(PDF)

**S4 Table. Input values of oral reference dose and cancer slope factor of each pesticide (USEPA, 2017).**
(PDF)

**S1 Graphical abstract.**
(TIF)

## Acknowledgments

Authors would like to acknowledge Chairperson Department of Zoology, Aligarh Muslim University, Aligarh for providing requisite facilities; simultaneously authors like to acknowledge USIF AMU, Aligarh for providing LCMS faculties to carry of this work.

## Author Contributions

**Conceptualization:** Zeshan Umar Shah, Saltanat Parveen.

**Data curation:** Zeshan Umar Shah, Saltanat Parveen.

**Formal analysis:** Zeshan Umar Shah.

**Investigation:** Zeshan Umar Shah, Saltanat Parveen.

**Methodology:** Zeshan Umar Shah, Saltanat Parveen.

**Resources:** Zeshan Umar Shah.

**Software:** Zeshan Umar Shah, Saltanat Parveen.

**Supervision:** Saltanat Parveen.

**Validation:** Zeshan Umar Shah, Saltanat Parveen.

**Visualization:** Zeshan Umar Shah, Saltanat Parveen.

**Writing – original draft:** Zeshan Umar Shah.

**Writing – review & editing:** Zeshan Umar Shah, Saltanat Parveen.

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
