## [Decision Letter · Decision Letter 0]

18 Aug 2022

PONE-D-22-19870Distribution and risk assessment of pesticide residues in sediment samples from river Ganga, IndiaPLOS ONE

Dear Dr. SHAH,

Thank you for submitting your manuscript to PLOS ONE. After careful consideration, we feel that it has merit but does not fully meet PLOS ONE’s publication criteria as it currently stands. Therefore, we invite you to submit a revised version of the manuscript that addresses the points raised during the review process.

The manuscript as a new submission still has many concerns after the evaluation by two reviewers.  The list of target chemicals is not shown in the Methods section, and quality assurance and control like LOQs and LODs for chemicals are unclear. The authors mentioned the LOQs and LODs for sediment are shown in their previous paper (Pesticide residues in Rita rita and Cyprinus carpio from river Ganga, India, and assessment of human health risk). However, after checking this paper, the information about LOQs and LODs for sediment is unavailable (only MDLs for water and fish there). The authors said that the pesticides were detected by the liquid chromatography equipped with a mass spectrometer, but they showed a GC column in the method section. Besides, this study must give convinced reasons why only pesticides in sediment were investigated, without consideration of their levels and risk in water. The manuscript must be thoroughly gone through before consideration for publication in the journal.

We look forward to receiving your revised manuscript.

Kind regards,

Guangjie Zhou

Academic Editor

PLOS ONE

Journal Requirements:

"No funding available to mention"

"No authors have competing interest"

4. Please amend the manuscript submission data (via Edit Submission) to include author Saltanat Parveen.

Reviewers' comments:

Reviewer's Responses to Questions

**Comments to the Author**

1. Is the manuscript technically sound, and do the data support the conclusions?

Reviewer #1: Partly

Reviewer #2: Partly

2. Has the statistical analysis been performed appropriately and rigorously? 

Reviewer #1: No

Reviewer #2: Yes

3. Have the authors made all data underlying the findings in their manuscript fully available?

Reviewer #1: No

Reviewer #2: Yes

4. Is the manuscript presented in an intelligible fashion and written in standard English?

Reviewer #1: Yes

Reviewer #2: No

5. Review Comments to the Author

Reviewer #1: The manuscript “Distribution and risk assessment of pesticide residues in sediment samples from river Ganga, India" has been re-submitted for publication in Plos One. In general, English phrasing has been reviewed improving the readability of the text. However, there are still several suggestions made by the reviewers that are not included in the text (for instance there is still a lack of information and explanations requested several times as the SPE, LOD, LOQs, etc.). Additionally, Table 2, Figures 2, 3, 4 and lines 199-213 present the same information. Supplementary Information is not explained and there are data missing. Consequently, manuscript still needs a MAJOR revision before being accepted for publication, which besides these comments considers the remarks highlighted in the attached document.

Reviewer #2: The authors report on the detection of 16 pesticide in sediment samples. The data from monitoring study was used to estimate the human health and ecotoxicological risk.

The introduction section is not informative enough. Aspects related to the risk posed by pesticides are not properly presented. Although the approach is focused on human health it includes major environmental risk factors which are not considered in a proper way.

The scope of the paper is not defined and the human health risk due to exposure from contaminated sediment is not justified.

The description of the analysis is not clear. The authors reported the use of LC-MS system for the detection of pesticide residues and the use of a GC column and ECD detection!!! Moreover, the operational conditions are not presented.

Moreover, it would be interesting to include a discussion section with data of currently used pesticides in the studied area and the legacy pesticides that have been detected.

6. PLOS authors have the option to publish the peer review history of their article (what does this mean?). If published, this will include your full peer review and any attached files.

Reviewer #1: No

Reviewer #2: No

---

## [Author Response · Author response to Decision Letter 0]

7 Nov 2022

Response to the reviewer’s comments:

Editor Comment: This study must give convinced reasons why only pesticides in sediment were investigated, without consideration of their levels and risk in water. 

Response: In our early research paper we have detected the level of pesticides in water from the studied area and there is lot of literature available on pesticide presence in water the river Ganga. Our main focus was to find out concentration of pesticides in sediment samples from the studied river and determination of probabilistic risk posed by the pesticides by calculating human and ecological risk. Few of the papers are mentioned below that have observed pesticides in water from river Ganga.

Divya, R., Ruby, P., Vikash, P., Sharma, P.K. and Shukla, D.N., 2014. Physico-chemical and pesticide analysis of River Ganga in Allahabad City, Uttar Pradesh, India. Asian Journal of Biochemical and Pharmaceutical Research, 4(3), pp.239-244.

Mutiyar, P.K. and Mittal, A.K., 2013. Status of organochlorine pesticides in Ganga river basin: anthropogenic or glacial?. Drinking Water Engineering and Science, 6(2), pp.69-80.

Leena, S., Choudhary, S.K. and Singh, P.K., 2011. Organochlorine and organophosphorous pesticides residues in water of River Gangaat Bhagalpur, Bihar, India. Int J Res Chem Environ, 1, pp.77-84.

Singh, L., Choudhary, S.K. and Singh, P.K., 2012. Pesticide concentration in water and sediment of River Ganga at selected sites in middle Ganga plain. International journal of environmental sciences, 3(1), pp.260-274.

Shah, Z.U. and Parveen, S., 2021. Pesticide residues in Rita rita and Cyprinus carpio from river Ganga, India, and assessment of human health risk. Toxicology Reports, 8, pp.1638-1644.

Samanta, S., 2013. Metal and pesticide pollution scenario in Ganga River system. Aquatic ecosystem health & management, 16(4), pp.454-464.

Reviewer 1:

1. Several suggestions made by the reviewers that are not included in the text (for instance there is still a lack of information and explanations requested several times as the SPE, LOD, LOQs, etc.

Response: SPE along with its methodology has been explained well in the section extraction of samples line number 112- 122. LOD, LOQ and recovery percentage of the detected pesticides has been well documented in the supplementary part (Table S2). The sediment samples were fortified with 10 μg/kg of the pesticides in sediments. Please refer to the supplementary part table S1.

2. Supplementary Information is not explained and there are data missing.

Response: In the supplementary part Table S1 gives few international studies on pesticide concentration detected in sediment samples as is explained in the manuscript line number 277-284, Table S2 gives the LOD, LOQ and Recovery percentage of each studied pesticide and is well explained in the quality control and quality assurance section line number 141-146, Table S3 in the supplementary file depicts input data from USEPA for the calculation of risk assessment, Table S4 gives the oral reference dose and cancer slop factor of each studied pesticides used for calculation of probable human health risk.

3. Table 2, Figures 2, 3, 4 and lines 199-213 present the same information. 

Response: Table 1 has been removed as figure 2, 3 and 4 present the same data as suggested by reviewer. Line number 199 – 213 explains the data well as it could not be properly explained in the figures.

4. Higher amounts of TOC were mentioned in middle and lower stretches, thus pesticide deposition was higher than upper stretch.

Response: No statistical support is present in this section; Total Organic Carbon was determined by partial oxidation method following Walkey and Black method as described by Trivedy and Goel (1998).

Reviewer 2:

1. Aspects related to the risk posed by pesticides are not properly presented.

Response: The section has been included in the introduction part in first paragraph line number 37-44.

2. The scope of the paper is not defined and the human health risk due to exposure from contaminated sediment is not justified.

Response: Line number 86-89, 306-319, 313-315, 31-40, 280-294 mainly focus on scope and human health risk due to exposure of contaminated sediments. 

3. The description of the analysis is not clear. The authors reported the use of LC-MS system for the detection of pesticide residues and the use of a GC column and ECD detection!!! Moreover, the operational conditions are not presented.

Response: the section has been reframed and operational conditions have been included in the section. 

4. It would be interesting to include a discussion section with data of currently used pesticides in the studied area and the legacy pesticides that have been detected.

Response: The section has been included in discussion part line number 243-250.

---

## [Decision Letter · Decision Letter 1]

20 Dec 2022

Distribution and risk assessment of pesticide residues in sediment samples from river Ganga, India

PONE-D-22-19870R1

Dear Dr. SHAH,

We’re pleased to inform you that your manuscript has been judged scientifically suitable for publication and will be formally accepted for publication once it meets all outstanding technical requirements.

Kind regards,

Guangjie Zhou

Academic Editor

PLOS ONE

Additional Editor Comments (optional):

Reviewers' comments:

Reviewer's Responses to Questions

**Comments to the Author**

1. If the authors have adequately addressed your comments raised in a previous round of review and you feel that this manuscript is now acceptable for publication, you may indicate that here to bypass the “Comments to the Author” section, enter your conflict of interest statement in the “Confidential to Editor” section, and submit your "Accept" recommendation.

Reviewer #2: All comments have been addressed

2. Is the manuscript technically sound, and do the data support the conclusions?

Reviewer #2: Yes

3. Has the statistical analysis been performed appropriately and rigorously? 

Reviewer #2: N/A

4. Have the authors made all data underlying the findings in their manuscript fully available?

Reviewer #2: Yes

5. Is the manuscript presented in an intelligible fashion and written in standard English?

Reviewer #2: Yes

6. Review Comments to the Author

Reviewer #2: Authors have adequate replied to all comments mentioned during the first review step. Moreover the additions made improve the final version of this manuscript. Thus I am suggesting the publication of this manuscript in its current form.

7. PLOS authors have the option to publish the peer review history of their article (what does this mean?). If published, this will include your full peer review and any attached files.

Reviewer #2: No

---

## [Editor Report · Acceptance letter]

24 Jan 2023

PONE-D-22-19870R1 

Distribution and risk assessment of pesticide residues in sediment samples from river Ganga, India 

Dear Dr. Shah:

I'm pleased to inform you that your manuscript has been deemed suitable for publication in PLOS ONE. Congratulations! Your manuscript is now with our production department. 

Kind regards, 

on behalf of

Dr. Guangjie Zhou 

Academic Editor

PLOS ONE